# Sexual Body Size Dimorphism in Small Mammals: A Case Study from Lithuania

**DOI:** 10.3390/biology13121032

**Published:** 2024-12-10

**Authors:** Linas Balčiauskas, Laima Balčiauskienė

**Affiliations:** Nature Research Centre, Akademijos 2, 08412 Vilnius, Lithuania; laima.balciauskiene@gamtc.lt

**Keywords:** voles, mice, shrews, age groups, morphometric differences, long-term research

## Abstract

We studied the differences in body size between males and females of 14 small mammal species in Lithuania. By measuring standard physical traits in voles, mice, and shrews from a large collection of long-term surveys, we updated information published 35 years ago and compared our results with data from other countries. Our results showed that males were larger than females in the yellow-necked mouse, root vole, and three other meadow voles, especially among adults. This pattern is consistent with Rensch’s rule, which refers to how size differences between the sexes change with total body size. In contrast, females were larger than males in bank voles and four mouse species, although this was less consistent across age groups. Shrews and the smallest mouse species, the herb field mouse, showed no significant size differences between sexes. In some species, these size differences changed as the animals grew. We are adding data on less common species such as the sibling vole and northern birch mouse, which have not been studied extensively. Our research provides new baseline data for small mammals in the boreal mid-latitudes and serves as a foundation for future studies of how animals adapt to changing ecological conditions or climate change.

## 1. Introduction

Sexual dimorphism is generally defined as differences between males and females of the same species [1]. These sex-related differences are not limited to sexual organs but can also affect growth rates, maturation, life expectancy, behavior [2], and certainly external traits. The term “sexual size dimorphism” (hereafter SSD) is used to analyze differences in body size with respect to animal sex [1].

Previous studies considered mammals to be potentially sexually dimorphic, with males being larger [3], and the driving force for this phenomenon was considered to be male–male competition [4]. However, there are a number of mammalian species with female-biased SSD, defined as 38 out of 186 species in [5]. This has been explained by higher energetic demands during pregnancy [6] and lactation [7], and by the fact that more body space is required to maintain eggs and embryos than to maintain sperm [4]. However, our knowledge of the physiological costs of reproduction in small mammals has been described as “rudimentary” [8].

If the size difference between males and females varies with total body size, this is an evolutionary regularity described as Rensch’s rule [9]. This rule suggests that, specifically in mammals, male body size tends to increase more rapidly than female body size in larger species. Therefore, in smaller species, males and females tend to be more similar in size, or females may even be slightly larger. In general, insectivores and rodents have been characterized as following Rensch’s rule, with males being larger [Lindenfors 2007], but other authors point out that not all small mammals follow Rensch’s rule [10].

In some mammalian taxa, such as mustelids, smaller species are more dimorphic [11], although in general larger species of the same genus are more dimorphic. In small mammals, this regularity is not clear. While some authors point to better expression in larger species [12], dimorphism is usually not expressed [13]. While in rodents, males are usually the larger sex [10], there are exceptions in microtines [12,14].

In mammals, data availability has limited the ability to estimate SSD [15]. In the absence of statistics in the literature, researchers have used arbitrary values of differences between means expressed as percentages [16] or simply relied on the male:female ratio being greater or less than 1 [17]. Therefore, SSD has been better studied in ungulates and pinnipeds [3,18,19], with a limited number of dedicated studies in rodents and other small mammals [14,20]. It is possible that relatively small sex differences in rodents have not attracted attention [12].

Data on dimorphism can be found in scientific papers not devoted to SSD. Data on the body weight of 10 small mammal species from Central Europe show that five species of mice, voles, and shrews are female-biased, three species are similar in weight, and two species are male-biased [21]. The results of a comprehensive study of SSD across a wide taxonomic range of rodent species are presented in [12], which reports monomorphism and SSD for 172 species based on structural characteristics—body length or body weight. However, this publication does not take into account possible geographic variation and limits the data to single populations. The most variable species in terms of SSD are the voles of the genera *Clethrionomys* and *Microtus*. In these genera, female-biased and male-biased SSDs are found even within the same species [12,20].

Recently, the availability of more comprehensive data has led to the refutation of earlier findings regarding male-biased dimorphism [15]. The dataset includes 429 mammalian species and presents the distribution of male-biased, female-biased, and monomorphic species across 17 mammalian orders. According to this review, the number of species categorized as female-biased, monomorphic, and male-biased were as follows: 4, 32, and 18 in the family Cricetidae; 4, 15, and 20 in the family Muridae; and 0, 7, and 6 in the family Soricidae. The family Sminthidae (birch mice) was not analyzed [15].

Long-term data are essential in the study of SSD in small mammals, as there are cases where the established pattern of SSD is reversed in the same species. In the yellow-necked mouse (*Apodemus flavicollis*), male-biased SSD has been reported in different regions of Europe [12,21,22,23]. In Lithuania, this species was reported as monomorphic [24]. However, in Slovakia, a case was recorded where the SSD changed from male-biased to female-biased in one of the six survey years, as the population responded to an exceptionally high population density. Therefore, the body size of the species was considered to be plastic in terms of adapting to changes in the short term [25]. This was explained by (i) a faster growth rate of females compared to males of similar size, and (ii) inter-annual variation in growth rates indicating a dependence on population density [26].

In Lithuania, previous research on male and female size differences in small mammals was summarized in the book Fauna of Lithuania: Mammals [24]. Although no statistics of the comparisons were presented, at least it is possible to recalculate if the SSD meets the 5% criterion formulated in [27], see Discussion. Furthermore, male and female body mass and body length were presented for vole, mouse, and shrew species in publications devoted to the analysis of differences in tooth row length between males and females and in different age groups. The main drawback is that the data for males and females were pooled, so the papers do not give a true picture of SSD [28,29,30]. Finally, for root vole (*Alexandromys oeconomus*), it was shown that SSD is best expressed in adult age [31], although the sample size was quite limited.

We aimed to test Lithuanian small mammals on the issue of SSD, using standard morphometric body measurements (body mass, body length, tail length, hind foot length, and ear length), and the consistency of these differences in age groups (juveniles, subadults, and adults). This is the first study of its kind based on long-term data and covering all species of voles, mice, and shrews in the same country, and, therefore, can claim to be a reference for boreal mid-latitude small mammals.

We believe that dividing the sample into six age-sex groups gives a better understanding of SSD in small mammals and shows how SSD is developed in postnatal development. This could explain the presence of two different patterns of body condition index in small mammals (either increasing or decreasing from juvenile to adult), as this index is calculated from body weight and length [32]. From a practical point of view, data on SSD should be a basis for pooling data on both sexes together or analyzing them separately. Our conclusions are based on the long-term experience of small mammal trapping in Lithuania, which has provided large samples of most species in the country.

## 2. Materials and Methods

### 2.1. Study Site and Small Mammal Trapping

The study site is Lithuania, a country located in northern Europe along the Baltic Sea, whose fauna and flora share similarities with other countries at similar latitudes, especially in northern and eastern Europe. It covers an area of 65,286 square kilometers and has mostly flat terrain with the highest elevation being about 294 m. The country’s climate is transitional between maritime and continental, with average temperatures ranging from −4.9 °C in January to 17.2 °C in July, and annual precipitation ranging from 570 to 902 mm, depending on location. Its vegetation includes mixed forests, coniferous and deciduous woodlands, meadows, pastures, and wetlands. Land use is divided into 57% agricultural land, 30% forests and shrubs, 4% water bodies, 3% swamps, and 6% other areas, supporting a population density of approximately 45.3 inhabitants per square kilometer.

For the analysis, we used data on small mammals caught in snap traps in the period 1975–2024 all over Lithuania (Figure 1). The main method used was trap lines with 25 traps, 5 m apart, set for three days and checked once or twice a day, i.e., in the morning or the morning and evening.

During this period, the trapping effort of 516,714 trap days resulted in the capture of 57,753 individuals of 20 small mammal species. Trapping occurred in 9 different habitats, including grassland, forest, wetland, shrub, commensal agricultural, disturbed, and riparian habitat, and if trapping occurred in several different habitats, then the habitat was designated as “mixed”. As already mentioned, trapping effort was not equal in all habitats [32].

### 2.2. Small Mammal Processing: Identification, Measurements, and Sample Size

No retrospective morphometric data are available for small mammal autopsies prior to the 1980s. In the 1980s–1990s, captured small mammals were processed in the field, while after the 2000s, captured individuals were kept frozen and transported to the laboratory for processing. Species were identified by external features, while *Microtus* voles were identified by differences in their teeth [24]. Genetic methods were used to identify sibling voles (*Microtus rossiaemeridionalis*) in the 1990s and 2023–2024. However, in most of the traps, we used *Microtus arvalis* sensu lato, as *M. rossiaemeridionalis* was not specifically identified. In addition, not all trapped individuals were measured for various reasons (body scraped, missing body parts, decomposition for autopsies, individuals not preserved for necropsy for a variety of reasons depending on the objectives of the study, etc.). Norway rat (*Rattus norvegicus*) and black rat (*R. rattus*) were not analyzed along with the other small mammals due to the difference in capture methods, the holding of individuals prior to necropsy, and the short collection period.

Three age groups (juveniles, subadults, and adults) were identified in necropsy in voles, mice, and shrews based on the development of their sexual organs [33,34], the degree of thymus atrophy [35], and only in a few cases on body mass. It has been shown that body mass varies greatly among age groups of small mammals and is, therefore, not the best indicator of age [36].

All non-breeding individuals with a well-developed thymus, filiform uterus in females, or testes in the abdominal cavity in males were defined as juveniles [37]. All non-breeding individuals with partially involuted thymus glands and inactive genitalia were defined as subadults [34]. All individuals with an involuted/atrophied thymus gland indicating reproductive activity (females: perforated or obstructed vagina, lactating, pregnant, corpora lutea in the ovaries [38] and/or placental scars in the uterus [39,40]; males: enlarged, scrotal testes, developed epididymis and seminal vesicles) or being post-reproductive were considered adults. Additional information on the identification of age groups is presented in [32].

Of the 29,203 individuals processed, 14,915 were males, 12,931 were females, and in four cases, both male and female sex organs were present in the same individual, so they were identified as hermaphrodites. The sex of 1353 individuals could not be determined. The sample composition of the dissected small mammals by species is shown in Table 1.

In four listed species, *N. milleri*, *M. avellanarius*, *A. sylvaticus*, and *A. amphibius*, the sample size was minimal, 10 individuals or less. For *N. fodiens*, *S. betulina*, *A. uralensis*, and *M. rossiaemeridionalis*, sample size was about one hundred individuals or less, so some age/sex groups are under-represented.

Animals were weighed and measured prior to necropsy. Body mass (Q) was measured to the nearest 0.1 g using spring, weighing, or electronic scales. Body length (L), tail length (C), hind foot length (P), and ear length (A) were measured to the nearest 0.1 mm using mechanical or electronic calipers in a standard manner [41]. Over 80% of the measurements were taken by the same person throughout the study period, minimizing potential bias according to [42]. However, not all of the above measurements were taken on all individuals captured, depending on the species and the purpose of the capture. For example, in small mammal monitoring, most trapped individuals were at least weighed and measured for body length, whereas in biodiversity research, only species, sex, and age group were recorded for the most abundant species. This resulted in different sample sizes for the measured parameters, with some species being underrepresented (Table 2).

### 2.3. Data Analysis

The normality of the distribution of Q, L, C, P, and A measurements was tested for each sex/age group with a sample size of at least 15 individuals using an online calculator with almost unlimited sample size [43]. We used the Shapiro–Wilk test, which is particularly suitable for small samples. Based on the *p*-value, we identified three groups of samples: those with normal distribution (*p* < 0.05), those with near-normal distribution (0.10 > *p* > 0.05), and non-normal samples (*p* > 0.10).

Fulfilling the condition that our data are continuous and the samples are randomly drawn from the species populations, we used ANOVA and a series of *t*-tests on all five traits (Q, L, C, P, and A) to find out if there is an SSD in the age groups of the analyzed small mammal species. According to [44], the *t*-test can be used with approximately normal distributions when the sample size is greater than 30 per group. The *t*-test is quite robust to mild deviations from normality, especially if the data do not have extreme outliers or highly skewed distributions, both of which are true for our data. For small samples, we used the Bayes factor (bootstrap with 9999 replications).

To test whether the reversal of dimorphism during postnatal development is significant, we used a linear mixed effects model with body mass (Q) as the dependent variable, sex and age groups, and their interaction, as fixed factors, capture site as a random factor, and month of capture as a covariate. If there was a significant interaction between sex and age group in determining body mass, reversal dimorphism was considered significant.

We also checked whether the difference between males and females reaches the 5% threshold by calculating it as (F − M)/F × 100, where F and M denote trait values in females and males, respectively. Negative values indicate larger trait sizes in males, positive values indicate larger trait sizes in females. In line with R.J. Smith’s criticism [45], we do not use the M/F ratio for our data or for comparisons with other authors’ samples.

Calculations were performed in Statistica for Windows, version 6.0 (StatSoft, Inc., Tulsa, OK, USA) [46] and in PAST version 4.13 (Museum of Paleontology, Oslo College, Oslo, Norway) [47]. The minimum confidence level was set as *p* < 0.05.

## 3. Results

### 3.1. Sexual Size Dimorphism in Lithuanian Small Mammal Species

Of the 18 small mammal species analyzed, the sample size was large enough for statistical analysis of male–female differences in 12 species, while in *S. betulina* and *M. rossiaemeridionalis*, the results may be dependent on the small sample size (see Table 1). Based on the significance of differences in five morphometric parameters, species were characterized as monomorphic or dimorphic, and in the latter case, with male-biased dimorphism or female-biased dimorphism (Table 3). The body mass variation in all the small mammal species studied is summarized in Appendix A (Table A1).

In addition, it was found that the statistical significance of the differences did not fully meet the 5% threshold—in several species where male–female differences were not significant, the 5% threshold of differences was met. These cases are discussed in the respective chapters.

### 3.2. Species with Male-Biased Sexual Size Dimorphism

Male-biased SSD was found in one mouse (*A. flavicollis*) and four vole species (*A. oeconomus*, *M. arvalis*, *M. agrestis*, and *M. rossiaemeridionalis*). Strong (greater than 5%) and significant male-biased SDD was characteristic of *A. oeconomus* (Figure 2) in all age groups, though not in all traits (Table 4). Significant, but of variable strength, male-biased SDD was characteristic of adult *A. flavicollis*. Ear length did not differ in other age groups of this species, and differences were less pronounced in juveniles (Table 4). Significant but of limited strength, not always reaching the 5% threshold, male-biased SSD was found in *M. arvalis*. Strong, exceeding the 5% threshold, but not significant SSD due to a smaller sample size was characteristic of *M. rossiaemeridionalis*, and male-biased SSD of limited strength and limited significance was found in *M. agrestis*. In the last three species, some parameters were female-biased, but neither strong nor significant, and only observed in immature individuals (Table 4).

In summary, consistent male-biased SSD in the adult age group was characteristic of all listed species and most characters. In subadults, some characters indicated female-biased SSD (Q and A in *M. arvalis*), while *A. flavicollis* and *M. rossiaemeridionalis* continued to show male-biased SSD. Juveniles of these species were the most variable in SSD. For example, *A. flavicollis* has more balanced dimorphism in juveniles, with some female-biased traits not strong, below the 5% threshold (Table 4). In Q, reversed dimorphism during postnatal development of this species was significant (F_2,6_ = 45.9, *p* < 0.0005).

### 3.3. Species with Female-Biased Sexual Size Dimorphism

Female-biased SSD was found in one vole and four mouse species, but not all of them were consistent in SSD expression across age groups and characters (Table 5). Adult individuals of the mentioned species show female-biased SSD in Q and L in all five species, in the consistently positive values. In two species, *M. minutus* and *S. betulina*, the differences in L were not significant. The highest female-biased SSD was a characteristic of adult individuals of *S. betulina*.

In non-adult animals of these species, SSD was inconsistent and less pronounced compared to adults, suggesting transitional growth patterns that vary among species. The five percent threshold was exceeded only for Q and A in *M. minutus*, and only the latter difference was significant.

The most variable SSD pattern was recorded in juveniles, with females of *S. betulina* and *M. musculus* being larger in Q, L, and C. In contrast, *A. agrarius*, *C. glareolus*, and *M. minutus* had slightly negative or neutral SSD in many parameters, suggesting that males and females are more similar in size at this stage. Thus, the juvenile age group showed the least consistent SSD among the species. In Q, reversed dimorphism during postnatal development of *A. agrarius* was significant (F_2,7_ = 15.7, *p* < 0.005). Considering Q in *M. minutus*, reverse dimorphism was strong but did not reach significance (F_2,2_ = 1.8, *p* = 0.11).

As an example of female-biased SSD, we present the distribution of morphological parameters in *A. agrarius* (Figure 3), which is present only in adult animals. A similar pattern is characteristic for adult *C. glareolus*—in both species, P was larger in males, by 0.8% and 0.4%, respectively, i.e., a small but significant difference (Table 5).

### 3.4. Monomorphic Small Mammal Species

The four small mammal species, three shrews and one mouse, could be classified as monomorphic with minimal SSD for most parameters and age groups (Table 6). Some parameters, such as ear length (A) in *S. minutus*, show higher female-biased SSD values in adults and subadults, suggesting unique developmental or ecological factors at play for this species. Juvenile stages show more variability, likely reflecting growth differences that stabilize by adulthood.

Adult individuals generally had a low SSD, with most values close to zero. The high female-biased SSD in the ear length (A) of *S. minutus* and *S. araneus* is atypical.

Subadults and juveniles of these species showed mixed results, with generally low SSD values, consistent with their monomorphic classification. The most variable in SSD was the juvenile age group, which showed a male-biased SSD, suggesting that male juveniles in this species grow faster and therefore reach certain sizes earlier than females.

As an example of a monomorphic species with inconsistent distribution of morphometric parameters in age groups, we present *S. araneus*, with some significant differences and also some exceeding the 5% threshold (Figure 4).

## 4. Discussion

Studies of the differences in physical characteristics between males and females of the same species, referred to as SSD, are important for several reasons [48]. The results help to understand species evolution and shaping factors [49]. Sexual dimorphism is related to the mating system of species, so understanding SSD provides insights into reproductive success, mate choice, and parental investment [50]. In small mammals, particularly microtines, the importance of SSD in two morphological traits, body mass and body length, in relation to the species mating system has been discussed, with mixed results [51,52]. In North American voles, species of *Microtus* have been shown to have male-biased SSD, while species of *Clethrionomys* have largely female-biased SSD [27]. Our results confirm this (see Table 4 and Table 5).

As an adaptation to different pressures, SSD may vary in different environments [53], with changes in male and female morphology along a north–south latitudinal gradient as shown for *A. uralensis* [54]. The relationship between SSD and trophic niche partitioning is well established in many different animal groups [49], with females selecting habitats and diets that differ significantly from those of males [1]. Competition for resources could lead to very rapid changes in SSD [55], although evidence for mammals is lacking. In extreme environments, such as Great Cormorant (*Phalacrocorax carbo*) breeding colony areas, the impact of avian activities affected the morphological parameters of *A. flavicollis* in less than a decade, confirming the possibility of rapid change under environmental pressure [56].

Three main factors in the evolution of SSD were identified: (i) correlation with body size, referred to as Rensch’s rule, (ii) differences in dietary niche, and (iii) mating system and social organization of the species [57]. However, the scientific debate on these factors is still open. For example, small mammals, insectivores, and rodents mostly have male-biased SSDs [4], although there are examples of subterranean rodents [10], bats [58], canids [59], and other animal groups [9] that do not follow Rensch’s rule.

There is a link between reproduction and SSD in small mammals, influenced by mating systems, ecological roles, reproductive investments, and environmental stressors. In flying squirrels (*Glaucomys volans*), larger females produce larger litters and have larger home ranges, allowing them to access more food resources [60]. A positive association between female body size and litter size has been found in many New and Old World voles [61], although such an association probably does not operate in shrews [62]. In Lithuania, we found positive correlations between female body mass and litter size in *A. flavicollis*, *A. agrarius*, *M. arvalis*, *M. agrestis*, *A. oeconomus*, and *C. glareolus* [63]. However, the correlation could be obscured by the fact that most small mammals continue to grow throughout their lives [64,65].

Therefore, as noted by [5], inconsistent statements about SSD in many species are at odds with the best available raw data. Our results, while consistent with the general conclusion that SSD is rarely observed in shrews [57], show male-biased SSD in one mouse and four vole species, and female-biased SSD in one vole and four mouse species. Therefore, we made comparisons with small mammal body size data from publications to see if our results (see Table 4, Table 5 and Table 6) were compatible with other countries at similar latitudes (Table 7).

Obviously, the classification in [12], where *M. agrestis* and *A. flavicollis* species were considered as male-biased SSD, *C. glareolus* as female-biased SSD, and *M. musculus* and *M. minutus* as monomorphic species, is not completely correct. As shown in Table 7, *M. minutus* had female-biased SSD and *A. flavicollis*, *M. agrestis*, and *M. arvalis* were male-biased. The remaining species (*S. araneus*, *S. minutus*, *N. fodiens*, *A*. *agrarius*, *A. uralensis*, *A. oeconomus*, and especially *C. glareolus*) show a wide variation from female-biased to male-biased SSD. Our results in Lithuania (see Table 4, Table 5 and Table 6) are in agreement with the data, discussion, and exclusions of other authors presented below.

In *Sorex* shrews, body mass is highly variable, while body length is poorly represented. In *S. araneus*, body mass varies from male-biased to female-biased values, with our data aligned at the higher end, but all values do not reach the 5% threshold. Body length is monomorphic in this species. Conversely, in *S. minutus*, the variation in body mass is substantial, from a male-biased to a female-biased SSD, exceeding the 10% threshold at both ends (Table 7). Our data show that this species in Lithuania was historically and currently monomorphic in body mass and length. Differences in other traits (Table 6) do not fit this picture and require additional temporal or spatial analysis. The third species, *N. fodiens*, should be considered monomorphic. Postnatal SSD changes in this shrew species are variable and require additional attention, as in our samples the proportion of individuals with unknown age or sex was high (see Table 1 and Table 2). The other peculiarity of this group is the reduction in body size in winter, a phenomenon known as the Dehnel effect, which helps to balance their high metabolic demands with environmental constraints [74]. This seasonal change involves a decrease in body mass, skull size, and some internal organs, with body size typically increasing again in the spring. The other problem is that shrews do not fit Bergmann’s rule, which states that larger body sizes are found in colder climates to conserve heat [75].

For *S. betulina*, our results show a strong female-biased SSD in both body mass and body length, in contrast to the monomorphic SSD suggested by [15]. The paucity of data in published sources and the comparatively small sample size in our analyses (see Table 2) should be noted. The species is not numerous in Lithuania and Poland [76], with even more fragmentation of the range in Western Europe [77]. Our data therefore contribute to the knowledge of species morphometry in general, not only for SSD.

As shown in [78,79], there is considerable variation in body size in *M. musculus*, and there does not appear to be a geographic cline in this species [24,65,66,67,68]. Sexual size dimorphism in *M. musculus* shows considerable variability across studies, ranging from a strong male bias to the strongest female bias reported in the older study from Lithuania [24]. Our data on body mass show a strong female-biased SSD, and in body length, fall within the range of previously reported values. In juveniles and subadults of this species, SSD was not expressed.

Body mass variation in *A. agrarius* shows a wide range across populations, from strongly male-biased SSD [69] to female-biased SSD, exceeding a 5% [24] or even 10% threshold (our data). Regarding body length, our results show that females are larger, in contrast to other authors [13,23,69]. Juveniles of *A. agrarius* had a male-biased SSD in body mass (−1.7%) and other measurements (see Table 5), suggesting a significant reversal of dimorphism during postnatal development.

Previous studies suggest a predominantly male-biased SSD in *A. flavicollis*, ranging from very strong [70,71] to neutral (Table 7). Our data are consistent with this male-biased trend and exceed the 10% threshold. Body length also shows a male-biased SSD, but the 5% threshold is not reached in any study. In subadult *A. flavicollis*, the male bias is less pronounced, but in juveniles of this species, body mass shows a female-biased SSD, indicating a statistically significant reversal of dimorphism at this stage. A case of reversion to female-biased SSD under exceptionally high population density was recorded in the West Tatra Mountains in Bulgaria [25].

Our data on *A. uralensis* indicate a moderate female-biased SSD, consistent with the upper range of previous findings, which ranged from −5.0% to 7.2% for body mass. The variation in body length SSD is much smaller. During postnatal development, both Q and L change from male-biased in juveniles and subadults to female-biased in adults. Variation in SSD was also found in previous studies, with no clear conclusion as to whether males or females were larger [23,80,81,82]. Male-biased SSD was characteristic of higher elevation localities [54].

Compared to the extreme value published previously [24], our finding of a strong and significant female-biased SSD in body mass of *M. minutus* and less than 5% SSD in body length is consistent with the other observations (Table 7), but not with the classification of the species as monomorphic by [12]. Juveniles of *M. minutus* in Lithuania were male-biased.

The widest range of SSD is found in *C. glareolus*, from strong male bias (−22.7%) to significant female bias (15.7%). Our data (8.1%) reflect a female-biased SSD within this spectrum for body mass and an insignificant female-biased SSD for body length. In postnatal development, changes in SSD occur with maturation, as juveniles and subadults were characterized by neutral SSD values (see Table 5).

The four meadow vole species (*A. oeconomus*, *M. agrestis*, *M. arvalis*, and *M. rossiaemeridionalis*) share the following characteristics in SSD: (i) all species exhibit male-biased SSD in adults, indicating that males tend to be larger than females for all traits examined, and (ii) in all these species, SSD changes during postnatal development, with juveniles and subadults being less male-biased.

The strongest male-biased SSD is characteristic of *A. oeconomus*, especially for body mass, which exceeds 15% in adults and 5% in other age classes. For body length, our data and those of [23] contrast with the report of female-biased SSD in [13]. In *M. agrestis*, male bias in body mass and body length, which does not reach the 5% threshold, contrasts with much stronger male bias reported by other authors (see Table 7), but is more consistent with previously reported monomorphism of this species in Lithuania [24]. In *M. arvalis*, where male-biased SSDs of varying strength (from −17.4% to −0.2%) have been reported, our results are in the range of the others. Finally, there are no data to compare strong male-biased SSD in *M. rossiaemeridionalis*, which we report. Males of this species were heavier than females in all age groups, especially in juveniles (see Table 4).

There are three methodological problems in studies of SSD in small mammals related to (i) age classification, (ii) the method used, and (iii) sample size.

Classification of age categories according to body mass, body length, fur condition, reproductive condition, and capture history in *C. glareolus* and *A. flavicollis* [36] defined *C. glareolus* less than 15 g and 85 mm and *A. flavicollis* less than 100 mm as juveniles. Based on reproductive parameters and thymus involution, we found 13 adults in *C. glareolus* with body mass less than 15 g and 69 adults with body length less than 80 mm. The same is true for *A. flavicollis*—the body mass of adults in our sample started from 15 g and the body length from 71 mm. The hibernation of small mammals observed in Lithuania [83] questions whether all hibernated individuals are adults [36,84], therefore, we assumed that reproductive parameters and thymus involution are the best parameters for age determination in small mammals.

Various methods were used to quantify SSD. This can be the slope of the reduced major axis of a regression of male size on female size, as recommended in [85], or a regression and residuals [45]. However, these methods require raw data, so regressions are rarely available for retrospective samples. Using a rate such as F/M on a linear scale eliminates the problem of spurious correlation or non-normality [45], leaving only the question of threshold. The 5% threshold [27] is a practical standard that helps researchers compare SSD across studies and assess its evolutionary and ecological implications. There are publications using lower thresholds for inferring differences, starting at 1% [86] and going as high as 10% [87]. Statistical significance in trait values between males and females based on *t*-tests, ANOVA, and post hoc has also been used, e.g., in [20,22,23]. In our study, we used a combination of ANOVA, *t*-tests, and F/M ratio with a 5% threshold for differences.

The sample sizes, which in our sample were larger than those cited, should be noted: *S. betulina* 11, *N. fodiens* 39, *A. sylvaticus* 20, and *A. uralensis* 33 individuals in [15], *A. oeconomus* 8, *A. agrarius* 18, and *M. minutus* 9 individuals in [13]. Therefore, our conclusions are better based on the actual data.

We illustrate the complexity of the SSD problem by considering a group of small mammals, mice of the genus *Apodemus*. In the broader ecological context, the comparison of *Peromyscus* and *Apodemus* as ecological equivalents is hampered by a lack of information, environmental heterogeneity, and methodological inconsistencies [88]. Nevertheless, many *Apodemus* species are considered to be dimorphic with larger males. Surprisingly, this is not always the case for the largest species, the Japanese field mouse (*Apodemus speciosus*), especially in isolated island populations [89,90,91]. Craniometric traits in *A. agrarius*, *A. sylvaticus*, and *A. flavicollis* from Bulgaria [92,93] not only confirmed the presence of SSD in these species but also emphasized the relevance of SSD in ecological and evolutionary contexts. Sex and body mass differences have also been linked to sexually dimorphic parasitism in *A. sylvaticus*, with potential implications for health and reproductive success [94].

Therefore, data on SSD in various small mammal species are important for advancing our understanding of rodent morphology, ecology, and evolutionary processes.

## 5. Conclusions

Our main results are the identification and numerical evaluation of male-biased SSD in *Apodemus flavicollis* and four vole species and female-biased SSD in *Clethrionomys glareolus* and four mouse species, and the characterization of adults as the age group with the most pronounced SSD. Several species showed transitions in dimorphism patterns during postnatal development. The strongest male-biased SSD was found in *Alexandromys oeconomus* and *Apodemus flavicollis*, the largest meadow vole and mouse species in the country, respectively, suggesting that these groups are subject to Rensch’s rule.

We present the evaluation of SSD in *Microtus rossiaemeridionalis* and *Sicista betulina*, the less common small mammals not analyzed by other authors. Furthermore, this study is a call for new analysis to determine whether there have been temporal changes in body size and SSD in Lithuanian small mammals during the last five decades that can be related to climate change.

Our analysis of SSDs for 14 species of small mammals, based on a large sample of long-term surveys, updated information published 35 years ago and showed that it is not advisable to rely on (especially older) SSDs from neighboring countries because of geographical or methodological differences.

## Figures and Tables

**Figure 1 biology-13-01032-f001:**
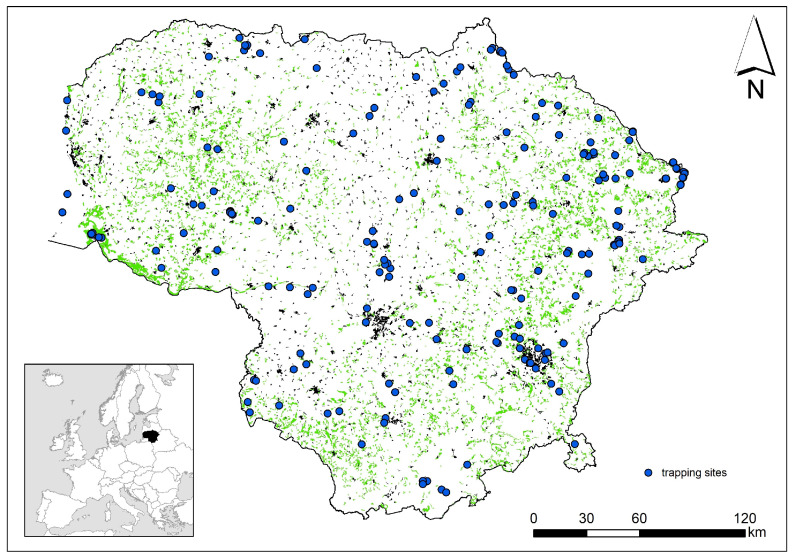
Trapping areas of small mammals in Lithuania, 1975–2024. Green color shows grasslands, black color shows urbanized areas. Lithuania’s position in Europe is shown in the inset map.

**Figure 2 biology-13-01032-f002:**
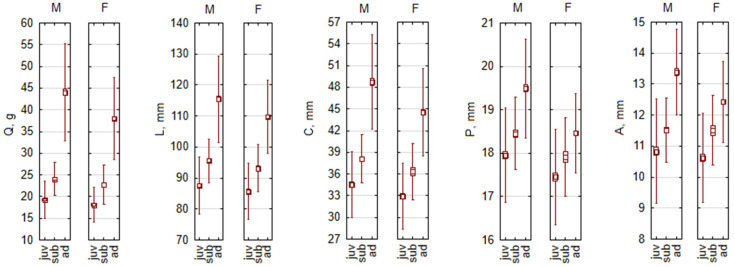
Male-based body size dimorphism by age groups in *Alexandromys oeconomus*, as best example of species with larger males. Abbreviations: M—males, F—females, juv—juveniles, sub—subadults, ad—adults, Q—body mass, L—body length, C—tail length, P—hind foot length, A—ear length.

**Figure 3 biology-13-01032-f003:**
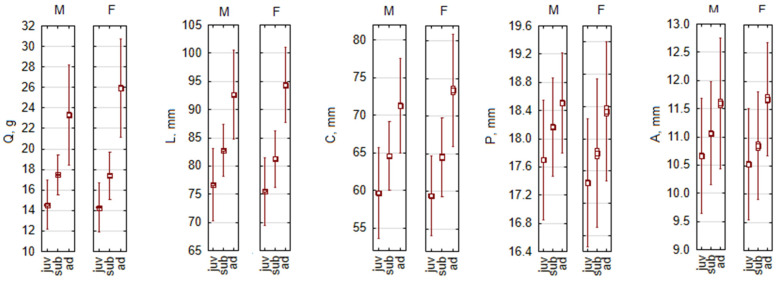
Female-based body size dimorphism by age groups in *Apodemus agrarius*, as an example of species with larger females. Abbreviations: M—males, F—females, juv—juveniles, sub—subadults, ad—adults, Q—body mass, L—body length, C—tail length, P—hind foot length, A—ear length.

**Figure 4 biology-13-01032-f004:**
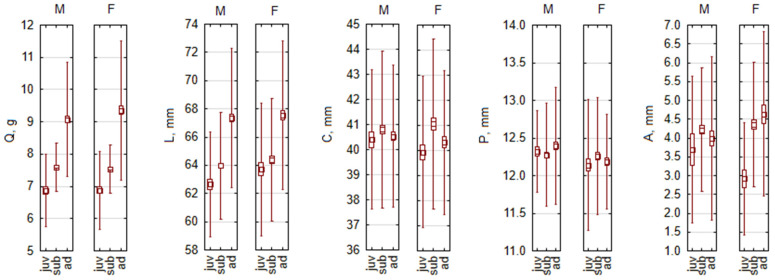
Sex-based body size dimorphism by age groups in *Sorex araneus*, as an example of monomorphic species. Abbreviations: M—males, F—females, juv—juveniles, sub—subadults, ad—adults, Q—body mass, L—body length, C—tail length, P—hind foot length, A—ear length.

**Table 1 biology-13-01032-t001:** Sample composition of dissected small mammals, 1980–2024, by sex and age.

Species	Adults		Subadults		Juveniles		Total
	Males	Females	Males	Females	Males	Females	
Common shrew (*Sorex araneus*) *	370	253	532	313	103	121	2549
Pygmy shrew (*Sorex minutus*) *	92	61	50	38	16	14	811
Water shrew (*Neomys fodiens*) *	21	18	13	13	5	0	101
Mediterranean water shrew (*Neomys milleri*) *	1	0	0	0	0	1	3
Hazel dormouse (*Muscardinus avellanarius*)	0	0	0	0	0	1	1
Northern birch mouse (*Sicista betulina*)	8	3	3	0	1	3	18
House mouse (*Mus musculus*)	110	74	97	24	70	65	440
Striped field mouse (*Apodemus agrarius*) *	456	333	755	331	966	931	3823
Yellow-necked mouse (*Apodemus flavicollis*) *	1585	1039	798	855	590	713	5666
Wood mouse (*Apodemus sylvaticus*)	2	0	0	1	2	0	5
Pygmy field mouse (*Apodemus uralensis*) *	4	16	9	14	14	16	74
Harvest mouse (*Micromys minutus*) *	18	40	60	34	108	78	350
Bank vole (*Clethrionomys glareolus*) *	1724	1796	2187	1329	1511	1759	10,398
Water vole (*Arvicola amphibius*)	0	1	2	2	5	0	10
Root vole (*Alexandromys oeconomus*) *	246	445	160	82	224	189	1356
Common vole (*Microtus arvalis*) *+	279	449	272	155	705	631	2510
Short-tailed vole (*Microtus agrestis*) *	117	210	135	58	120	77	719
Sibling vole (*Microtus rossiaemeridionalis*)	20	34	4	3	3	2	66

*—The total number of individuals captured (N) and the sum of identified age and gender numbers differ; +—sensu lato. In most studies, *M. rossiaemeridionalis* was not specifically identified.

**Table 2 biology-13-01032-t002:** Sample composition of weighed and measured small mammals by species. Abbreviations: Q—body mass, L—body length, C—tail length, P—hind foot length, A—ear length.

Species	Number of Individuals						
	Age	Gender	Q	L	C	P	A
*Sorex araneus*	2256	1871	2416	2321	1572	1565	1021
*S. minutus*	326	468	765	732	420	433	253
*Neomy fodiens*	93	78	100	100	64	92	72
*N. milleri*	2	3	3	3	1	2	2
*Muscardinus avellanarius*	1	1	1	1	1	1	1
*Sicista betulina*	18	18	17	17	17	17	16
*Mus musculus*	440	440	438	432	224	231	224
*Apodemus agrarius*	3785	3778	3578	3531	2012	2104	2053
*A. flavicollis*	5585	5576	5551	5508	3403	3522	3463
*A. sylvaticus*	5	5	5	4	4	5	4
*A. uralensis*	74	73	69	83	72	74	72
*Micromys minutus*	346	338	343	341	265	275	271
*Clethrionomys glareolus*	10,341	10,317	10,158	10,015	6701	6602	6089
*Arvicola amphibius*	10	10	10	10	10	10	10
*Alexandromys oeconomus*	1350	1349	1340	1308	974	984	943
*Microtus arvalis* *	2491	2487	2460	2432	983	987	948
*M. agrestis*	718	718	714	698	459	459	439
*M. rossiaemeridionalis*	66	66	66	66	29	30	30

*—sensu lato. In most studies, *M. rossiaemeridionalis* was not specifically identified.

**Table 3 biology-13-01032-t003:** Age-related sexual size dimorphism in small mammal species from Lithuania based on five main morphological parameters. Significant differences indicated in bold type. Abbreviations: M—males, F—females, Q—body mass, L—body length, C—tail length, P—hind foot length, A—ear length.

Species	Age Group	Q	L	C	P	A
*Sorex araneus*	Adults	F > M	M = F	M = F	**M > F**	**F > M**
	Subadults	M = F	M = F	M = F	M = F	M = F
	Juveniles	M = F	F > M	M = F	**M > F**	M > F
*S. minutus*	Adults	M = F	M = F	M = F	M = F	M = F
	Subadults	M = F	F > M	M = F	M = F	**F > M**
	Juveniles	M = F	M = F	M = F	M = F	M = F
*Neomys fodiens*	Adults	M = F	M = F	**M > F**	M = F	M = F
	Subadults	M = F	M = F	M = F	M = F	M = F
	Juveniles	Males only				
*Sicista betulina*	Adults	M = F	F > M	M = F	M = F	M = F
	Subadults	Males only				
	Juveniles	M = F	M = F	M = F	M = F	M = F
*Mus musculus*	Adults	**F > M**	**F > M**	M = F	M = F	M = F
	Subadults	M = F	M = F	M = F	M = F	M = F
	Juveniles	M = F	M = F	M = F	M = F	M = F
* Apodemus agrarius*	Adults	**F > M**	**F > M**	**F > M**	**M > F**	M = F
	Subadults	M = F	**M > F**	M = F	**M > F**	**M > F**
	Juveniles	**F > M**	**F > M**	M = F	**F > M**	**F > M**
* A. flavicollis*	Adults	**M > F**	**M > F**	**M > F**	**M > F**	**M > F**
	Subadults	**M > F**	**M > F**	**M > F**	**M > F**	M = F
	Juveniles	F > M	**F > M**	**F > M**	**M > F**	M = F
* A. uralensis*	Adults	M = F	M = F	M = F	**M > F**	M = F
	Subadults	M = F	M = F	M = F	M = F	M = F
	Juveniles	M = F	M = F	M = F	M = F	M = F
* Micromys minutus*	Adults	**F > M**	M = F	M = F	M = F	F > M
	Subadults	M = F	F > M	M = F	M = F	**F > M**
	Juveniles	M = F	M = F	M > F	M = F	M = F
* Clethrionomys glareolus*	Adults	**F > M**	**F > M**	**F > M**	**M > F**	**F > M**
	Subadults	M = F	M = F	M = F	**M > F**	M = F
	Juveniles	M = F	M = F	M = F	M > F	**M > F**
* Alexandromys oeconomus*	Adults	**M > F**	**M > F**	**M > F**	**M > F**	**M > F**
	Subadults	**M > F**	**M > F**	**M > F**	**M > F**	M = F
	Juveniles	**M > F**	M > F	**M > F**	**M > F**	M = F
* Microtus agrestis*	Adults	M = F	**M > F**	M > F	**M > F**	M = F
	Subadults	M = F	M = F	M = F	M > F	M = F
	Juveniles	M = F	M = F	M = F	M = F	M = F
* M. arvalis*	Adults	**M > F**	**M > F**	**M > F**	**M > F**	**M > F**
	Subadults	F > M	**F > M**	M = F	**M > F**	**F > M**
	Juveniles	**M > F**	**M > F**	**M > F**	**M > F**	**M > F**
* M. rossiaemeridionalis*	Adults	M > F	M = F	M = F	M = F	M = F
	Subadults	M > F	M = F	M = F	M = F	M = F
	Juveniles	M = F	M = F	M = F	M = F	M = F

M = F—monomorphic species, *p* > 0.10 (shaded green); M > F—male-biased traits, *p* < 0.10 (no shading); F > M—female-biased traits, *p* < 0.10 (no shading); **M > F**—male-biased traits, *p* < 0.05 (light blue shading); **F > M**—female-biased traits, *p* < 0.05 (light orange shading); **M > F**—male-biased traits, *p* < 0.01 (medium blue shading); **F > M**—female-biased traits, *p* < 0.01 (medium orange shading); **M > F**—male-biased traits, *p* < 0.001 (dark blue shading); **F > M**—female-biased traits, *p* < 0.001 (dark orange shading).

**Table 4 biology-13-01032-t004:** Percentage expression of sexual size dimorphism in male-biased small mammal species. Negative values indicate larger size in males, positive values indicate larger size in females. Abbreviations: Q—body mass, L—body length, C—tail length, P—hind foot length, A—ear length. Bold indicates reliable differences with *p* < 0.05.

Age Group	Parameter	*A. flavicollis*	*A. oeconomus*	*M. arvalis*	*M. rossiaemeridionalis*	*M. agrestis*
Adults	Q	**−11.2**	**−15.8**	**−5.0**	−12.2	−3.8
	L	**−3.8**	**−5.2**	**−2.6**	−3.6	−2.9
	C	**−2.2**	**−9.4**	−3.0	−3.4	−3.0
	P	**−3.4**	**−5.6**	**−3.4**	−2.1	**−2.7**
	A	**−2.9**	**−7.8**	**−2.8**	−2.1	−1.2
Subadults	Q	**−8.9**	**−5.9**	3.0	−15.5	0.1
	L	**−2.4**	**−2.6**	**1.7**	−3.6	−0.7
	C	**−1.4**	**−4.7**	1.2	5.5	1.8
	P	**−1.8**	**−3.0**	**−1.8**	−4.9	−2.3
	A	0.7	0.0	**6.3**	−7.4	−1.7
Juveniles	Q	2.8	**−6.3**	**−6.1**	−40.3	−2.9
	L	**1.5**	−2.1	**−1.2**	−21.1	0.4
	C	**1.9**	**−5.1**	**−5.1**	6.2	−3.0
	P	**−0.8**	**−2.9**	**−3.0**	−0.2	−0.2
	A	0.7	−2.0	**−3.6**	−20.5	0.8

**Table 5 biology-13-01032-t005:** Percentage expression of sexual size dimorphism in female-biased small mammal species. Positive values indicate larger size in females, negative values indicate larger size in males. Abbreviations: Q—body mass, L—body length, C—tail length, P—hind foot length, A—ear length. Bold indicates reliable differences with *p* < 0.05.

Age Group	Parameter	*A. agrarius*	*C. glareolus*	*M. minutus*	*M. musculus*	*S. betulina*
Adults	Q	**10.1**	**8.1**	**14.2**	**13.2**	17.9
	L	**1.8**	**2.2**	1.5	**4.7**	13.3
	C	**3.0**	**5.4**	1.9	1.9	−3.8
	P	**−0.9**	**−0.4**	−1.3	−1.0	4.3
	A	0.7	**1.8**	5.4	2.2	8.8
Subadults	Q	−0.5	0.1	5.4	0.1	
	L	**−1.8**	0.0	2.8	−2.3	
	C	0.0	−0.5	2.6	−1.5	
	P	**−2.0**	**−0.6**	0.4	−1.3	
	A	**−1.9**	−0.3	**10.6**	−4.1	
Juveniles	Q	**−1.7**	0.2	−2.7	2.2	27.6
	L	**−1.6**	−0.1	−1.2	2.8	3.2
	C	−0.5	−0.2	−2.5	3.2	5.5
	P	**−1.4**	−0.4	−0.1	−0.3	−4.8
	A	**−1.3**	**−1.2**	−2.9	1.9	−9.1

**Table 6 biology-13-01032-t006:** Percentage expression of sexual size dimorphism in monomorphic small mammal species. Positive values indicate larger size in females, negative values indicate larger size in males. Abbreviations: Q—body mass, L—body length, C—tail length, P—hind foot length, A—ear length. Bold indicates reliable differences with *p* < 0.05.

Age Group	Parameter	*S. araneus*	*S. minutus*	*N. fodiens*	*A. uralensis*
Adults	Q	3.1	0.9	−1.9	4.5
	L	0.3	1.3	0.0	1.4
	C	−0.5	0.9	**−6.4**	−1.3
	P	**−1.3**	0.2	−1.7	**−7.8**
	A	**13.8**	31.9	−4.9	−1.3
Subadults	Q	−0.9	3.5	0.8	−4.6
	L	0.6	2.9	1.0	−0.8
	C	0.4	−1.6	−0.6	0.8
	P	−0.3	0.4	2.3	−2.9
	A	3.2	**58.1**	−3.0	−1.7
Juveniles	Q	0.1	−10.6		−3.9
	L	1.7	0.9		−0.6
	C	−1.2	−2.0		3.3
	P	**−1.9**	−2.8		−0.4
	A	−26.5	5.7		1.1

**Table 7 biology-13-01032-t007:** Variation in percentage expression of sexual size dimorphism (SSD) in European small mammals. Positive values indicate larger character size in females, while negative values indicate larger character size in males. Our data show values of adult individuals. Abbreviations: Q—body mass, L—body length.

Species	Trait	SSD Expression (%) [Source]	Our Data
*Sorex araneus*	Q	−3.0 [15], 0.0 [24], 4.6 [15], 4.7 [21]	3.1
	L	0.0 [24]	0.3
*S. minutus*	Q	−15.6 [21], 0.0 [24], 0.3 [15], 10.8 [15]	0.9
	L	0.0 [24]	1.3
*Neomys fodiens*	Q	0.0 [24], 0.1 [15], 2.1 [15], 3.4 [21]	−0.9
	L	0.0 [24]	0.0
*Sicista betulina*	Q	1.2 [15]	17.9
	L		13.3
*Mus musculus*	Q	−19.6 [66], −18.7 [67], −11.8 [65], −9.8 [65], −9.4 [65], 4.5 [68], 7.1 [68], 24.2 [24]	13.2
	L	−3.2 [15], −1.8 [12], 3.1 [13], 5.7 [24]	4.7
*Apodemus agrarius*	Q	−21.0 [69], −6.7 [23], −2.4 [23], 1.5 [21], 1.8 [15], 7.2 [24]	10.1
	L	−7.4 [69], −4.0 [23], −2.4 [23], −0.8 [13], 1.2 [24]	1.8
*A. flavicollis*	Q	−17.7 [70], −16.6 [71], −8.6 [21], −7.0 [23], −6.6 [23], −6.2 [15], −6.2 [22], −1.5 [22], 0.0 [24]	−11.2
	L	−4.4 [71], −3.3 [12], −2.0 [23], −1.8 [23], 0.0 [24]	−3.8
*A. uralensis*	Q	−5.0 [23], −1.1 [54], 3.4 [21], 6.9 [15], 7.2 [21]	4.5
	L	−1.7 [54]. −1.6 [23], 1.0 [23]	1.4
*Micromys minutus*	Q	1.6 [15], 15.7 [13], 31.0 [24]	14.2
	L	0.2 [12], 7.8 [24]	1.5
*Clethrionomys glareolus*	Q	−22.7 [20], −19.5 [20], −18.3 [20], −15.0 [20], −11.7 [20], −9.6 [20], −8.6 [20], −5.4 [20], −4.8 [20], −3.7 [20], −3.1 [20], −2.7 [20], −0.9 [20], −0.7 [20], 0.0 [20], 0.8 [20], 1.6 [20], 2.9 [15], 4.4 [21], 5.5 [20], 5.8 [61], 6.9 [20], 8.3 [20], 9.1 [15], 10.0 [12], 10.0 [15], 10.0 [20], 11.2 [24], 15.6 [15], 15.7 [15]	8.1
	L	3.4 [24],	2.2
*Alexandromys oeconomus*	Q	−28.6 [15], −25.0 [61], −24.0 [15], −14.9 [72], −8.5 [73], 3.5 [73]	−15.8
	L	−3.1 [73], −3.5 [73], 2.3 [13]	−5.2
*Microtus agrestis*	Q	−29.0 [15], −22.2 [15], −16.2 [61], −15.1 [15], −15.1 [72], 0.0 [24]	−3.8
	L	0.0 [24]	−2.9
*M. arvalis*	Q	−17.4 [72], −2.7 [24], −0.8 [21], −0.2 [61]	−5.0
	L	−0.8 [24]	−2.6
*M rossiaemeridionalis*	Q		−12.2
	L		−3.6

References and comments: [13]—small sample size; [15]—different populations; [23]—adults and subadults presented separately; [24]—when stating that males and females do not differ in size and providing pooled data, we arbitrarily set the percentage to zero; [54]—pooled data from most of the range; [61]—alpine populations, three sites sampled for several years in spring and autumn, recalculated; [65]—30–35 days old individuals, geographically distant populations; [71]—male bias also for C, P and A; [73]—adults and subadults presented separately, male-biased also for C, P, and A.

## Data Availability

This is an ongoing research, therefore, data are available from the corresponding author upon request.

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
