# Peer review of "Sexual Body Size Dimorphism in Small Mammals: A Case Study from Lithuania"

_biology, 2024, doi:10.3390/biology13121032_

Round 1
Reviewer 1 Report
Comments and Suggestions for Authors
In this paper, the authors assess sexual size dimorphism in 14 species of small mammals in Lithuania, citing ambiguity in previously published literature and a need to strengthen available data as justificiations for the research. Overall, I find the publication acceptable. The sample sizes, overall, are good. Although some species did not have enough individuals to draw SSD results, I think its important to get the data out there.
This brings me to my first comment: I think it would be really useful for the authors to make the raw data available for future studies. The authors cite that there is ongoing research. At the very least, however, the authors should provide a Supplemental table that has male and female averages and standard deviations for the various species/age groups. This would make the work more citeable and allow others to compare to their data.
Overall, the paper is well-written. I did, however, find the species by species discussion after Table 7 a little tedious. While acceptable, I think a more concise, synthesized discussion of the overview of differences with published literature could be more impactful. Individuals have the table to refer to for the details, and you’ve already discussed individual species in the above sections. Getting lost in the details kind of took my attention away to the overall points.
Is Table 7 pesenting only adult values? I assume so, but please make clear. Personally, the adult values were the most informative to me. I imagine that juvenile and subadult values could vary based on the actual age distribution of individuals within those samples. I don't believe this was brought up in the discussion.
Is there any data about how much these species adult body masses fluctuate throughout the year (based on season)? I also found myself wondering whether that variation could contribute to the ambiguous and contradictory results. In general, when dealing with small mammals, small fluctuations can have a bigger impact on percent differences. I think seasons were brought up briefly, but this point again gets lost. Revising those paragraphs after Table 7 to focus on these major points instead of species by species comparisons could really help highlight these issues and possible contributing factors to contradictory info.
Author Response
Rev#1 comments and answers
Comment: In this paper, the authors assess sexual size dimorphism in 14 species of small mammals in Lithuania, citing ambiguity in previously published literature and a need to strengthen available data as justificiations for the research. Overall, I find the publication acceptable. The sample sizes, overall, are good. Although some species did not have enough individuals to draw SSD results, I think its important to get the data out there.
Answer: exactly as you say here, some species are under-represented, but they are under-represented also in other countries. We were hesitating to include species with small sample size, however, even these data might be of value.
Comment: This brings me to my first comment: I think it would be really useful for the authors to make the raw data available for future studies. The authors cite that there is ongoing research. At the very least, however, the authors should provide a Supplemental table that has male and female averages and standard deviations for the various species/age groups. This would make the work more citeable and allow others to compare to their data.
Answer: as for the raw data policy, (i) we have legal binding for some raw data not to make available, as they were collected under various projects, and we been only allowed to use data for scientific purposes. This single reason is essential not to open our dataset. We expect not making the raw data publicly available is not detrimental to the quality of this paper.
Finally, we plan several more cooperative publications, already receiving data from other countries, this again binds us not to open raw data as a part of agreements.
As for averages and SD – all these are included into figures. 14 species, 2 sex and 3 age categories in each, plus 5 traits – this is 420 cells, a bit too much for a one Table, especially when this information is already presented graphically. However, we are answering your comment by including the main trait, body mass, in a Table format as an Appendix.
Comment: Overall, the paper is well-written. I did, however, find the species by species discussion after Table 7 a little tedious. While acceptable, I think a more concise, synthesized discussion of the overview of differences with published literature could be more impactful. Individuals have the table to refer to for the details, and you’ve already discussed individual species in the above sections. Getting lost in the details kind of took my attention away to the overall points.
Answer: We cannot agree without reservations, even though we initially thought exactly as you suggest. Therefore, in Lines362–369, a summary and correspondence to our result is given.
However, when examining the data presented in the cited publications in detail, we found several papers on changes in body mass and length that are incorrectly estimated because the original data used are outside any known limits. This is the subject of our next article, not this one you review. We have therefore decided that a broader discussion of the species is necessary for the reader to understand, without looking at the primary biases, whether we are making accurate generalizations.
Comment: Is Table 7 pesenting only adult values? I assume so, but please make clear. Personally, the adult values were the most informative to me. I imagine that juvenile and subadult values could vary based on the actual age distribution of individuals within those samples. I don't believe this was brought up in the discussion.
Answer: We also assume so, if not stated exactly in the cited publications. These details are in the footnote of the Table 7. However, not all publications are so detailed.
Comment: Is there any data about how much these species adult body masses fluctuate throughout the year (based on season)? I also found myself wondering whether that variation could contribute to the ambiguous and contradictory results. In general, when dealing with small mammals, small fluctuations can have a bigger impact on percent differences. I think seasons were brought up briefly, but this point again gets lost. Revising those paragraphs after Table 7 to focus on these major points instead of species by species comparisons could really help highlight these issues and possible contributing factors to contradictory info.
Answer: data used in Table 7 are not sufficient for such analysis, and this was not the aim of the first, most general study on the data from Lithuania. Our data would allow seasonal analysis, at least for the most abundant species. Thank you for idea, we will realize it in the future.
Reviewer 2 Report
Comments and Suggestions for Authors
The manuscript entitled “Sexual body size dimorphism in small mammals: a case study from Lithuania” aims to evaluate SSD in several species of small mammals from Lithuania using statistical analyses (ANOVA, t-tests, thresholds, etc.) and a long-term database compiled from 1975 to 2024. This research holds significant interest due to the central role of body size and mass in the complex network of interdependent biological variables. Consequently, the manuscript’s topic is highly appealing to zoologists. The extensive dataset provides a solid foundation for obtaining robust and reliable results, and the statistical methods employed align well with the research objectives. The manuscript is well-written and well-organized. The introduction effectively covers the key concepts related to body mass, SSD and Rensch's rule, making it accessible and beneficial for non-specialist readers in the future. Additionally, the discussion is well-structured and coherent. The authors have successfully achieved the primary goal of their research, and I would like to congratulate them on producing such an interesting and excellent work.
I do not have major concerns regarding this study. However, I am attaching a line-by-line review (PDF) with suggestions, details, questions, and comments for the authors to consider. The most important points are as follows:
- Provide more details about certain decisions mentioned in the Materials and Methods section. For example: “not all trapped individuals were measured for various reasons,” the exclusion of Norway rats and black rats, and clarification regarding the ratio of differences between sexes and their significance (how is it calculated?).
- Be clearer in the tables and some text sections. See comments in the pdf.
- Some data, such as Shapiro-Wilk test results, are not provided. I encourage the authors to include these in supplementary files.
- The boxplots in the figures appear misaligned; please check and correct this issue (lines of the box are not consistent).
- Question: Have you evaluated potential SSD differences between populations (of the same species) from different environments? You mentioned sampling across various habitats, but it is unclear whether differences in SSD across these habitats have been analyzed.
I have also noticed some misspellings and grammatical errors that the authors should review before publication.
I hope to see this manuscript published soon.
Best regards,

Author Response
Rev#2 comments and answers
Comment: The manuscript entitled “Sexual body size dimorphism in small mammals: a case study from Lithuania” aims to evaluate SSD in several species of small mammals from Lithuania using statistical analyses (ANOVA, t-tests, thresholds, etc.) and a long-term database compiled from 1975 to 2024. This research holds significant interest due to the central role of body size and mass in the complex network of interdependent biological variables. Consequently, the manuscript’s topic is highly appealing to zoologists. The extensive dataset provides a solid foundation for obtaining robust and reliable results, and the statistical methods employed align well with the research objectives. The manuscript is well-written and well-organized. The introduction effectively covers the key concepts related to body mass, SSD and Rensch's rule, making it accessible and beneficial for non-specialist readers in the future. Additionally, the discussion is well-structured and coherent. The authors have successfully achieved the primary goal of their research, and I would like to congratulate them on producing such an interesting and excellent work.
Answer: thank you
I do not have major concerns regarding this study. However, I am attaching a line-by-line review (PDF) with suggestions, details, questions, and comments for the authors to consider. The most important points are as follows:
Comment: Provide more details about certain decisions mentioned in the Materials and Methods section. For example: “not all trapped individuals were measured for various reasons,” the exclusion of Norway rats and black rats, and clarification regarding the ratio of differences between sexes and their significance (how is it calculated?)
Answer: acknowledged and explained in the text. In short: significance of differences difference used t-test and ANOVA, 5% is as threshold for magnitude (formula given in revision). While at small sample even 15% difference might be not significant, big sample support significant differences with magnitude less than 1%. Therefore, both are required.
Comment: Be clearer in the tables and some text sections. See comments in the pdf.
Answer: we accepted your comments sent in pdf and did necessary changes
Comment: Some data, such as Shapiro-Wilk test results, are not provided. I encourage the authors to include these in supplementary files.
Answer: We appreciate the reviewer’s suggestion to include Shapiro-Wilk test results for all parameters in supplementary files. However, given the complexity of our dataset—with 14 species, 2 sex categories, 3 age categories, and 5 parameters per species—providing a detailed breakdown of normality tests for every parameter would result in an exceptionally large supplementary table or file, potentially overwhelming readers without adding substantial value.
Additionally, as stated in the manuscript, we reported that not all parameters were normally distributed, and this information has been incorporated into our analytical framework to ensure appropriate statistical methods were applied. We used t-test, as according Statistics Knowledge Portal it can be used with approximately normal distributions when the sample size is greater than 30 per group. The t-test is quite robust to mild deviations from normality, especially if the data do not have extreme outliers or highly skewed distributions, both of which are true for our data. For small samples, we used the Bayes factor (bootstrap with 9999 replications). And, t-test was just one of the two methods for SSD evaluation – we also user 5% criterion.
Comment: The boxplots in the figures appear misaligned; please check and correct this issue (lines of the box are not consistent)
Answer: apologies, at a moment we choose not to make changes in the Figures, as these all alignment come directly from the program Statistica and based on our data. Any change to align central point to be visible would be tampering, in our understanding, as it violates two principles, namely
(1) Accurate Representation: graphs must be presented truthfully without manipulation that could mislead readers or distort results, and
(2) No Alterations: modifying elements like axes, scales, or data points to enhance visual appeal or reinforce conclusions.
Eg., Cromey DW. Digital images are data: and should be treated as such. Methods Mol Biol. 2013;931:1-27. doi: 10.1007/978-1-62703-056-4_1.
Or
Appropriate Image Manipulation in Scientific Publishing, https://www.letpub.com/Appropriate_Image_Manipulation_in_Scientific_Publishing
Question: Have you evaluated potential SSD differences between populations (of the same species) from different environments? You mentioned sampling across various habitats, but it is unclear whether differences in SSD across these habitats have been analyzed.
Answer: no, it was not in the aims of publication.
Comment: I have also noticed some misspellings and grammatical errors that the authors should review before publication.
Answer: pdf based comments on misspellings and errors were all answered, thank you for the careful reading. However,
- We leave “male-based dimorphism” in Line 76 as is, as we think this differs from SSD abbreviation.
- We use “inset map” in the caption of Figure 1, as a standard term in cartography, indicating a smaller map or graphical representation included within the main map; it serves to zoomed-out context, in this case providing a broader context – the location of the main map's area within a larger region
- Comments for Lines 230–234 are answered above
- Line 243 – no, we present exact values, even if these are less than 5% - depending on the sample size, these can be significant
- Line 244 – SSD by definition means male-females size difference, and this is shown in the table. Age groups are given, in the first column.
- Line 322: no, we do not analyze habitat influence.
Comment Line 369: Please indicate which of these changes are significant and which not. For example, in some parragraphs you talk about "reversal of dimorphism".. but it it significant? If not, we cannot be sure about this reversion.
Answer: statistical comparisons with published data are not possible, as there are no required data present in the publications cited. However, we fully acknowledge your comment of the significance of reversed SSD with our dataset.
In revision, we used linear mixed-effects model on the body mass (Q) as dependent variable, sex and age groups, as fixed factors, plus their interaction, trapping location as random factor, and month of trapping as covariate. If there is a significant interaction between sex and age group in determining body mass, reversal dimorphism is considered significant.
This way, significant reversal of dimorphism during postnatal development was confirmed in body mass of A. flavicollis and A. agrarius. We incorporated additional text in respective parts of the manuscript.
Reviewer 3 Report
Comments and Suggestions for Authors
Very well designed and performed study showing in details the variability among different species of small rodents. Perhaps a short paragraph in the discussion could be added, explaining for readers outside of Europe the environmental characteristics of Lithuania and their potential implications for the development of small mammals populations.
Author Response
Rev#3 comments and answers
Comment: Very well designed and performed study showing in details the variability among different species of small rodents. Perhaps a short paragraph in the discussion could be added, explaining for readers outside of Europe the environmental characteristics of Lithuania and their potential implications for the development of small mammals populations.
Answer: thank you, indeed country characteristic was missing. We changed first part of Material and Methods to 2.1. Study Site and Small Mammal Trapping, ad added required information. We think it is better to characterize site in the beginning, not in Discussion.
The study site is Lithuania, a country located in northern Europe along the Baltic Sea, whose fauna and flora share similarities with other countries at similar latitudes, especially in northern and eastern Europe. It covers an area of 65,286 square kilometers and has mostly flat terrain with the highest elevation is about 294 meters. The country's climate is transitional between maritime and continental, with average temperatures ranging from –4.9°C in January to 17.2°C in July, and annual precipitation ranging from 570 to 902 mm, depending on location. Its vegetation includes mixed forests, coniferous and deciduous woodlands, meadows, pastures, and wetlands. Land use is divided into 57% agricultural land, 30% forests and shrubs, 4% water bodies, 3% swamps, and 6% other areas, supporting a population density of approximately 45.3 inhabitants per square kilometer.